# Hypocalcemia on Admission Is a Predictor of Disease Progression in COVID-19 Patients with Cirrhosis: A Multicenter Study in Hungary

**DOI:** 10.3390/biomedicines11061541

**Published:** 2023-05-26

**Authors:** Bálint Drácz, Veronika Müller, István Takács, Krisztina Hagymási, Elek Dinya, Pál Miheller, Attila Szijártó, Klára Werling

**Affiliations:** 1Department of Surgery, Transplantation and Gastroenterology, Semmelweis University, 1083 Budapest, Hungary; hagymasi.krisztina@med.semmelweis-univ.hu (K.H.); miheller.pal@med.semmelweis-univ.hu (P.M.); szijarto.attila@med.semmelweis-univ.hu (A.S.); werling.klara@med.semmelweis-univ.hu (K.W.); 2Department of Pulmonology, Semmelweis University, 1083 Budapest, Hungary; muller.veronika@semmelweis.hu; 3Department of Internal Medicine and Oncology, Semmelweis University, 1083 Budapest, Hungary; takacs.istvan@med.semmelweis-univ.hu; 4Digital Health Department, Semmelweis University, 1083 Budapest, Hungary; dinya.elek@public.semmelweis-univ.hu

**Keywords:** liver cirrhosis, COVID-19, hypocalcemia, severity, mortality

## Abstract

Hypocalcemia is a common condition in liver cirrhosis and is associated with the severity of SARS-CoV-2 infection. However, there is a lack of data demonstrating the prognostic value of hypocalcemia in COVID-19 patients with cirrhosis. This study aimed to evaluate the prognostic value of hypocalcemia for COVID-19 severity, mortality and its associations with abnormal liver function parameters. We selected 451 COVID-19 patients in this retrospective study and compared the laboratory findings of 52 COVID-19 patients with cirrhosis to those of 399 COVID-19 patients without cirrhosis. Laboratory tests measuring albumin-corrected total serum calcium were performed on admission, and the levels were monitored during hospitalization. The total serum calcium levels were significantly lower in cirrhosis cases (2.16 mmol/L) compared to those without cirrhosis (2.32 mmol/L). Multivariate analysis showed that hypocalcemia in COVID-19 patients with cirrhosis was a significant predictor of in-hospital mortality, with an OR of 4.871 (*p* < 0.05; 95% CI 1.566–15.146). ROC analysis showed the AUC value of total serum calcium was 0.818 (95% CI 0.683–0.953, *p* < 0.05), with a sensitivity of 88.3% and a specificity of 75%. The total serum calcium levels showed a significant negative correlation with the Child–Turcette–Pugh score (r = −0.400, *p* < 0.05). Hypocalcemia on admission was a significant prognostic factor of disease progression in COVID-19 patients with cirrhosis.

## 1. Introduction

SARS-CoV-2 infection caused by severe acute respiratory syndrome coronavirus 2 (SARS-CoV-2) has spread across the world, emerging as a global health crisis. Data have shown that COVID-19 is associated with the risk of cirrhosis progression and poor outcomes [1,2,3]. Cirrhosis-associated immune dysfunction (CAID) is a common phenomenon in patients with cirrhosis. CAID comprises alterations in both innate and acquired immunity, leading to systemic inflammation and immunodeficiency [4]. Increased gut permeability and intestinal dysbiosis are also observed. Furthermore, this condition is associated with increased bacterial translocation and an increased prevalence of bacterial infections [5]. Therefore, patients with cirrhosis are more vulnerable to SARS-CoV-2 infection, and the early identification of risk factors is fundamental. Large international registry cohorts showed that overall mortality increased progressively with the severity of liver cirrhosis [6,7]. Patients with decompensated cirrhosis following SARS-CoV-2 infection were found to be at an increased risk of poor outcomes [8]. Liver cirrhosis is usually associated with hypocalcemia due to hypoalbuminemia and vitamin D deficiency [9,10]. In immune cells, calcium signals control a wide range of functions, such as differentiation, maturation, phagocytosis and cytokine secretion [11]. The change in cytosolic calcium concentration might play an essential role in the distortion of the host cell immune system, which benefits pathogen survival and virulence [12]. Several studies have shown the relation between the level of serum calcium and disease severity in SARS-CoV-2 infection. Luigi et al. reported that hypocalcemia can be a risk factor for intensive care unit (ICU) admission and worse prognosis in COVID-19 patients [13]. Hypocalcemia was highly prevalent in hospitalized patients following SARS-CoV-2 infection and thereby contributed to the increased occurrence of oxygen support and higher mortality rates [14,15]. There are limited data about the prognostic value of hypocalcemia in COVID-19 patients with cirrhosis.

Therefore, we aimed to investigate the prognostic value of hypocalcemia for disease severity and mortality in COVID-19 patients with cirrhosis. Moreover, the aim of our study was to determine the degree of association between the levels of albumin-corrected total serum calcium and abnormal liver function parameters in cirrhosis.

## 2. Materials and Methods

### 2.1. Study Population

Between March 2020 and May 2022, 6394 COVID-19 patients were retrospectively recruited in several centers of Semmelweis University. The diagnosis of SARS-CoV-2 infection was confirmed through a positive reverse transcription polymerase chain reaction (RT-PCR, SEQONCE qPCR Multi Kit, IVD, SeqOnce Biosciences, Carlsbad, CA 92008, USA) test based on a nasopharyngeal swab using the protocol of the World Health Organization [16]. High-resolution computer tomography (HRCT, Philips Incisive 128, Philips, Amsterdam, the Netherlands) was performed for all the included patients for a prompt diagnosis of SARS-CoV-2 infection. Liver cirrhosis was previously diagnosed based on radiomorphological alterations (e.g., ascites, liver surface nodularity), clinical characteristics of portal hypertension (e.g., gastrointestinal varices on endoscopy), liver elastography and liver biopsy. We selected 451 SARS-CoV-2-confirmed inpatients aged ≥ 18 years with elevated liver transaminases (>40 U/L) on admission or liver cirrhosis in their medical histories. Regardless of COVID-19 severity and the presence of pneumonia, the patients were divided into 2 groups: 52 COVID-19 patients with cirrhosis and 399 COVID-19 patients without cirrhosis. Of the 6394 COVID-19 patients, 626 COVID-19 patients aged < 18 years were excluded from our study. Furthermore, we excluded 5317 COVID-19 patients with normal levels of liver enzymes or the absence of liver disease on admission (Figure 1).

### 2.2. Study Design

We performed a retrospective multicenter study using data from electronic medical records. All data of the confirmed COVID-19 patients, including clinical characteristics and laboratory findings, were recorded and reviewed. Our study protocol was approved by the Scientific and Research Ethics Committee of the Medical Research Council of Hungary (IV/7946-1/2021/EKU, Budapest, 14 October 2021). It conformed to the ethical norms and standards of the Declaration of Helsinki.

On admission, a physical assessment was conducted, and a detailed, comprehensive patient history was recorded with special emphasis on the clinical course and severity of liver cirrhosis.

The concept of stratification and diagnostic criteria for COVID-19 severity were published based on national protocol in accordance with the 7th version of the guidelines established by the Chinese National Health Commission [17].

### 2.3. Data Collection

Laboratory examinations were consistently performed at the onset of SARS-CoV-2 infection and followed until discharge or death. The criteria for the COVID-19 patients’ discharge from hospital were as follows: (1) the resolution of fever for >48 h without antipyretics and (2) without supplementary oxygen therapy, (3) no signs of increased work in breathing or respiratory distress, (4) improvement in the signs and symptoms of illness (cough, shortness of breath, and oxygen requirement), and (5) two consecutive negative RT-PCR tests based on nasopharyngeal swabs, taken at least 24 h apart. During the hospital stay, laboratory biomarkers were measured regularly, including liver transaminases (AST, ALT), cholestatic parameters (GGT, total bilirubin, direct bilirubin, ALP), liver function tests (albumin, INR, total protein), inflammatory biomarkers (CRP), a complete blood count and basic metabolic panel (sodium, potassium, total serum calcium, glucose, creatinine, glomerular filtration rate).

The total serum calcium concentration in patients with hypoalbuminemia may not accurately reflect the physiologically active calcium concentration [18]. Therefore, we calculated the level of calcium corrected for albumin using the following correction formula: corrected calcium = (0.8 × [normal albumin − patient’s albumin]) + serum calcium [19]. A corrected serum calcium level below 2.2 mmol/L (8.9 mg/dL) was considered as hypocalcemia [20].

### 2.4. Cirrhosis Severity Grades

The modified Child–Turcette–Pugh (CTP) score and Model for End-Stage Liver Disease sodium (MELD-Na) score were applied to grade the severity of cirrhosis. The CTP score employs the following five clinical measures: total serum bilirubin, serum albumin, the international normalized ratio (INR), ascites grades and the stages of hepatic encephalopathy. The clinical classification of the CTP score is as follows: 5 to 6 points are considered as class A (well-compensated cirrhosis), 7 to 9 points are class B (significant functional compromise) and 10 to 15 points are class C (decompensated cirrhosis) [21]. The interpretation of ascites was determined based on the amount of abdominal fluid: grade 1 ascites detected via ultrasound; grade 2 ascites defined based on relative abdominal distension; and grade 3 ascites with compelling abdominal extension [22]. The West Haven Criteria were used for grading the severity of hepatic encephalopathy [23]. The European Association for the Study of the Liver Chronic Liver Failure Consortium (EASL-CLIF-C) criterion was applied for predicting prognosis in hospitalized cirrhotic patients with acute decompensation [24]. Acute-on-chronic liver failure (ACLF) was diagnosed according to the EASL-CLIF-C and the North American Consortium for the Study of End-Stage Liver Disease (NACSELD) criteria [25,26]. The MELD-Na score was calculated using values based on serum bilirubin, serum creatinine, INR, serum sodium and hemodialysis treatments obtained at least twice in the past week to predict the survival probability [27].

### 2.5. Statistical Analysis

Statistical analysis was performed using SPSS software version 28 (IBM Corporation, Armonk, NY, USA). The Kolmogorov–Smirnov test was used for testing the normality of all the data. The data were found to be non-normally distributed. Categorical variables were presented as frequencies and percentages. The results of continuous data were characterized using descriptive statistics: the number of samples (*n*) and median with the interquartile range. Pearson’s chi-square test and the two-tailed Fisher’s exact test were applied for a comparison of the categorical variables of COVID-19 patients with cirrhosis to those without cirrhosis. Continuous variables were compared with the Mann–Whitney U test to determine significance between the two groups. The abnormal laboratory markers were measured using univariate and multivariate logistic regression. Odds ratios (OR) with 95% confidence intervals (CI) were calculated. We performed a receiver operating characteristic (ROC) curve analysis to investigate the capacity of sodium (Na), serum calcium corrected for albumin, albumin, INR, white blood cell (WBC) and CTP levels to predict mortality. The Youden index was used to calculate the cut-off values. Spearman’s rank correlation coefficient (Spearman’s r) was calculated for comparisons between risk factors and disease progression. The analysis was two-sided with a significance level of α = 0.05.

## 3. Results

### 3.1. Baseline Characteristics and Laboratory Findings

The baseline characteristics of COVID-19 patients with cirrhosis (*n* = 52) vs. COVID-19 patients without cirrhosis (*n* = 399) are compared in Table 1. The median age of the COVID-19 patients with cirrhosis was significantly lower compared to those without cirrhosis (62 years vs. 65 years). COVID-19 patients with cirrhosis had longer hospital stays (14 days vs. 11 days, *p* = 0.082). Nevertheless, the mortality rates of COVID-19 patients without cirrhosis vs. COVID-19 patients with cirrhosis were 11.8% and 9.6%, respectively. Regarding minerals, the median values of potassium, albumin-corrected total serum calcium and magnesium were significantly lower in the cirrhosis cases (*p* < 0.05). Kidney function laboratory tests, including eGFR and serum creatinine, showed a significant difference between the two groups (*p* < 0.05). Regarding the inflammatory markers and complete blood count, the median values of the red blood count, platelets and CRP were significantly lower in COVID-19 patients with cirrhosis compared to those without cirrhosis *p* < 0.05). By contrast, the median value of the white blood cell (WBC) count was higher in the cirrhosis patients.

### 3.2. Etiology and Severity of Liver Cirrhosis

The common causes of cirrhosis are depicted in Figure 2. Among the 52 cirrhosis cases, alcohol was the most typical cause (27/52). Among the viral etiological agents, hepatitis C virus (HCV) and hepatitis B virus (HBV) were common, being present in 15.4% (8/52) and 3.8% (2/52) of the patients, respectively. Regarding autoimmune liver diseases, autoimmune hepatitis (AIH), primary biliary cholangitis (PBC) and primary sclerosing cholangitis (PSC) occurred in 9.6% (5/52), 3.8% (2/52) and 11.5% (6/52) of cirrhosis patients, respectively. The remaining two cirrhosis cases were cryptogen (3.8%).

All COVID-19 patients with cirrhosis were classified into three stages of severity, as follows: compensated cirrhosis, 7.7% (4/52); decompensated cirrhosis, 80.8% (42/52); and ACLF, 11.5% (6/52). Most of the ascites grades were severe (35/52). Regarding encephalopathy, stage 1–2, stage 3 and stage 4 encephalopathy occurred in 48% (25/52), 34.6% (18/52) and 17.3% (9/52) of cases, respectively (Figure 3).

### 3.3. Hypocalcemia as a Significant Predictor of In-Hospital Mortality in COVID-19 Patients with Cirrhosis

As demonstrated in Table 2, age and hospital stay were independently associated with fatal outcomes in COVID-19 patients with cirrhosis. In the univariate analysis, albumin, INR, total bilirubin, direct bilirubin and CTP were closely associated with in-hospital mortality. Furthermore, Na, WBC and platelets were significant prognostic factors for poor outcomes, with ORs of 0.905 (*p* < 0.05; 95% CI 0.847–0.966), 1.314 (*p* < 0.05; 95% CI 1.226–1.409) and 0.995 (*p* < 0.05 95% CI 0.990–1.000), respectively.

In a multivariate logistic regression assessing mortality based on gender and cirrhosis severity, Na, albumin, INR, direct bilirubin, WBC and CTP remained significant risk factors for mortality in COVID-19 patients with cirrhosis. Moreover, hypocalcemia was independently associated with in-hospital mortality, with an OR of 4.871 (*p* < 0.05; 95% CI 1.566–15.146)

### 3.4. Predictive Value of Total Serum Calcium Levels in Deceased COVID-19 Patients with Cirrhosis

The ROC curves of sodium, total serum calcium, albumin, INR, WBC and CTP are depicted in Figure 4. For total serum calcium, the area under the curve (AUC) value was 0.818 (95% CI 0.683–0.953, *p* < 0.05), which was the second highest among the factors evaluated. As demonstrated in Table 3, the optimal cut-off value was 2.02 mmol/L, with a sensitivity of 88.3% and a specificity of 75%, which were the highest among the factors investigated.

### 3.5. Hypocalcemia on Admission Is Closely Associated with Disease Severity in COVID-19 Patients with Cirrhosis

The COVID-19 patients with cirrhosis classified according to different total serum calcium levels are compared in Table 4. Patients with hypocalcemia on admission had higher mortality rates compared to normocalcemic patients (16% vs. 3.7%, *p* > 0.05). As illustrated in Table 4, most cirrhosis cases were decompensated in both groups. With respect to decompensation events, severe ascites occurred more frequently in hypocalcemic patients (21/25) relative to normocalcemic patients (14/27). Regarding COVID-19 severity, patients with hypocalcemia had severe respiratory failure requiring mechanical ventilation during hospitalization. However, oxygen therapy was common, being used in 55.6% of normocalcemic patients. Hypocalcemic patients were older and required prolonged hospitalization compared to normocalcemic patients (65 vs. 60, 14 vs. 13). The median values of direct bilirubin, GGT and CTP were significantly higher in the hypocalcemic patients relative to patients with normocalcemia (*p* < 0.05). Nevertheless, the hypocalcemic patients were closely associated with lower levels of albumin and total protein (*p* < 0.05).

An albumin-corrected total serum calcium level below 2.2 mmol/L (8.9 mg/dL) was considered as hypocalcemia. An albumin-corrected total serum calcium level between 2.2 mmol/L and 2.6 mmol/L was considered as normocalcemia [28].

### 3.6. Correlation between Hypocalcemia and Disease Severity in COVID-19 Patients with Cirrhosis

The results of Spearman’s correlation analysis are depicted in Table 5. Albumin-corrected total serum calcium showed significant negative correlations with direct bilirubin, GGT and CRP: r = −0.275, r = −0.350 and r = −0.341, respectively. As illustrated in Figure 5, there was a significant negative correlation between the total serum calcium level and CTP (r = −0.400, *p* < 0.05).

## 4. Discussion

Hypocalcemia is a common feature of liver cirrhosis and highly prevalent in SARS-CoV-2 infection. Therefore, we conducted a multicenter retrospective study to evaluate the prognostic value of hypocalcemia for disease progression and mortality in COVID-19 patients with cirrhosis.

Paizis et al. revealed that the ACE2 receptor is abundantly expressed in cirrhotic versus healthy livers, suggesting that patients with cirrhosis are at increased risk of adverse outcomes. [29]. In our analysis, we found that COVID-19 patients with cirrhosis required prolonged hospitalization. In line with previous studies, COVID-19 patients with cirrhosis had higher levels of liver transaminases and liver function parameters and lower albumin levels on admission [30,31,32]. Relevant to cirrhosis, we found that COVID-19 patients were more prone to the development of significant thrombocytopenia and hypocalcemia and worsening kidney function on admission [13,33,34].

Relevant to the etiology of cirrhosis, we identified alcohol use disorder as the most frequent cause (Figure 2). Patients with alcohol abuse are vulnerable to infections principally owing to their immune dysfunction and poor general health. During the pandemic, the medical condition of these patients might have worsened due to alcohol relapse and postponed medical checkups.

There are certain clinical manifestations of cirrhosis in COVID-19 patients. Hepatic decompensation events including worsening ascites or hepatic encephalopathy are prevalent, being present in up to 46% of patients [35]. Our data also showed that COVID-19 was a trigger for acute hepatic decompensation events, mostly with severe ascites (Figure 3). A multinational cohort study including 29 countries and a multicenter matched-cohort study in Hungary reported that a stepwise increase in cirrhosis severity classified based on the CTP score was associated with higher mortality rates in SARS-CoV-2 infection [35,36]. We also found that the CTP score was a reliable predictor of mortality in cirrhosis patients following SARS-CoV-2 infection. Previous studies showed that higher levels of total bilirubin, direct bilirubin and INR and lower albumin levels are risk factors for disease progression and poor prognosis in SARS-CoV-2 infection [37,38,39,40]. According to the data, higher levels of total bilirubin, direct bilirubin and INR and lower albumin levels were significant predictors of in-hospital mortality.

In a systemic review and meta-analysis investigating 953 COVID-19 patients, hypocalcemia was associated with poor outcomes, with a sensitivity of 76% and specificity of 53% [41]. In an analysis of the performance and predictive value of prognostic factors for mortality, the total serum calcium corrected for albumin level was found to be a highly sensitive and specific predictive marker for mortality. The cut-off value of total serum calcium is considered as hypocalcemia (Table 3). Previous studies reported that hypocalcemia could be used to predict poor prognosis, with an AUC value above 0.70 [41,42]. Regarding its diagnostic accuracy, hypocalcemia was a highly efficient predictive marker in our study, with an AUC value of 0.818 (95% CI 0.683–0.953, *p* < 0.05) (Table 3). Our data suggest that total serum calcium corrected for albumin is an accurate and highly sensitive prognostic factor that can be used to predict mortality in COVID-19 patients with cirrhosis (Figure 4).

In a meta-analysis of 2032 patients, hypocalcemia was associated with disease severity and fatal outcomes in COVID-19 patients [43]. In a retrospective study comparing the clinical course of COVID-19 patients with hypocalcemia to that of patients with normocalcemia, the hypocalcemic patients were associated with severe disease and prolonged hospitalization [44]. According to our findings, hypocalcemia on admission was a significant predictor of mortality in COVID-19 patients with cirrhosis. Although mineral deficiencies, including hypocalcemia, are common in liver cirrhosis, the patients with hypocalcemia were more vulnerable to acute decompensation following SARS-CoV-2 infection (Table 4) [45,46]. Our study also revealed that ascites grading was closely associated with lower calcium levels, suggesting that hypocalcemia is a clinical warning of disease progression in COVID-19 (Table 4).

Several studies have reported that SARS-CoV-2 virus can use calcium for replication, and an excessive immune response to SARS-CoV-2 infection might impair calcium homeostasis [47,48]. In our analysis, the concentration of albumin-corrected serum calcium was inversely proportional to the level of CRP and, consequently, the severity of disease (Figure 5). As demonstrated in Table 4, worsening respiratory status was associated with decreased calcium levels. The increased occurrence of mechanical ventilation in hypocalcemic patients might serve as a clinical warning indicating the exacerbation of SARS-CoV-2 infection. The findings of this study also confirmed that worsening hypocalcemia can imply the dysregulation of the immune response, leading to disease progression and the increased occurrence of decompensation events. These findings of the present study are consistent with the findings of Alemzadeh [49]. 

Altogether, our data provide evidence showing that hypocalcemia on admission can act as a reliable predictor of disease progression in COVID-19 patients with cirrhosis. Cirrhosis patients with more severe hypocalcemia might show deterioration in the cirrhosis stage and have a more excessive immune response following SARS-CoV-2 infection. Therefore, albumin-corrected total serum calcium is of specific importance, as it can serve as a red flag indicating disease progression in cirrhosis patients following SARS-CoV-2 infection.

The limitations of our study include its retrospective study design and relatively low number of patients. First, the number of COVID-19 patients with liver cirrhosis was limited in our analysis. The major hospital outcomes were restricted to in-hospital mortality and the duration of hospitalization. The criteria for discharge, including two negative RT-PCR tests obtained consecutively at least 24 h apart, could have influenced the major outcomes. The higher mortality rates of COVID-19 patients without cirrhosis compared to those with cirrhosis support the notion that patients with cirrhosis show higher adherence to regular monitoring and surveillance programs, which prevent severe illness and death following SARS-CoV-2 infection. Second, the laboratory parameters including serum 25-hydroxyvitamin D [25(OH)D], parathyroid hormone (PTH) and the arterial blood gas test were excluded owing to the unavailability of laboratory tests. Therefore, further prospective studies should investigate the prognostic value of hypocalcemia in association with serum 25-hydroxyvitamin D [25(OH)D], parathyroid hormone (PTH) and blood gas levels.

On the other hand, the strengths of our study include its multicenter design and its scope, comprising a patient group highly vulnerable to COVID-19-related adverse outcomes. Although there are numerous international studies investigating the impact of hypocalcemia on the clinical outcomes of SARS-CoV-2 infection, our study uniquely provides a better understanding of the prognostic role of hypocalcemia in cirrhosis patients following SARS-CoV-2 infection.

## 5. Conclusions

Hypocalcemia on admission was a significant prognostic factor of disease progression and severity in COVID-19 patients with cirrhosis. Cirrhosis patients with hypocalcemia were highly prone to excessive immune response and deterioration in terms of the cirrhosis stage. Since it is feasible to measure serum calcium in emergency departments, serum calcium levels should be employed to assess the severity of disease in COVID-19 patients with cirrhosis.

## Figures and Tables

**Figure 1 biomedicines-11-01541-f001:**
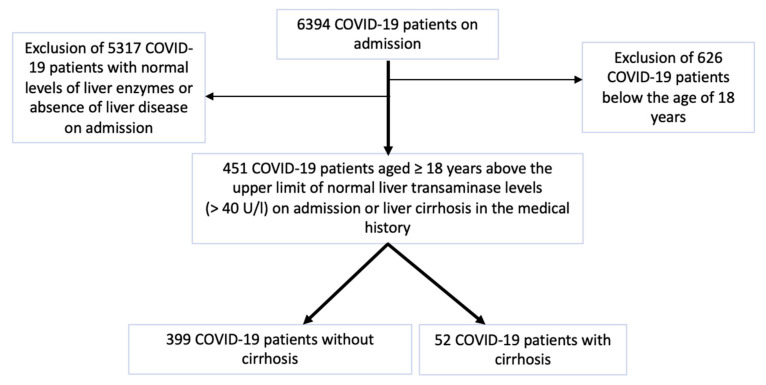
Flow chart of patient selection.

**Figure 2 biomedicines-11-01541-f002:**
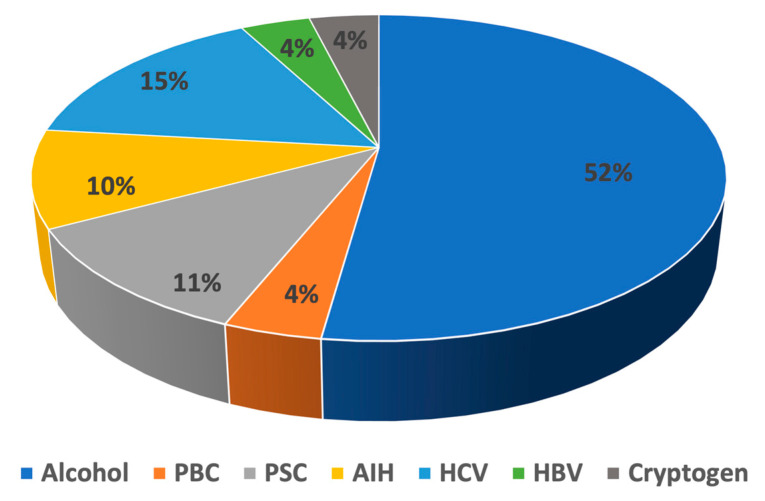
Distribution of common causes in COVID-19 patients with cirrhosis. PBC, primary biliary cholangitis; PSC, primary sclerosing cholangitis; AIH, autoimmune hepatitis; HCV, hepatitis C virus; HBV, hepatitis B virus.

**Figure 3 biomedicines-11-01541-f003:**
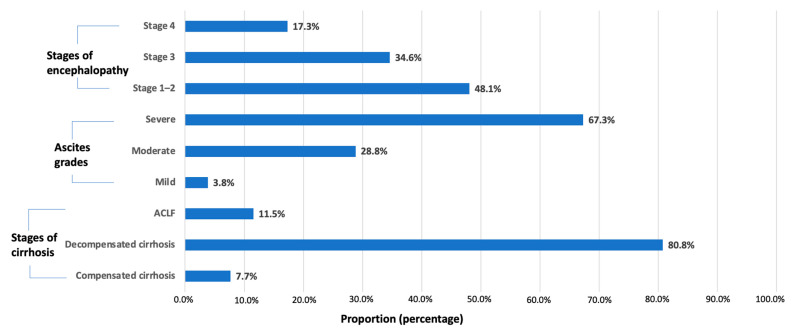
Clinical features of cirrhosis severity in COVID-19 patients.

**Figure 4 biomedicines-11-01541-f004:**
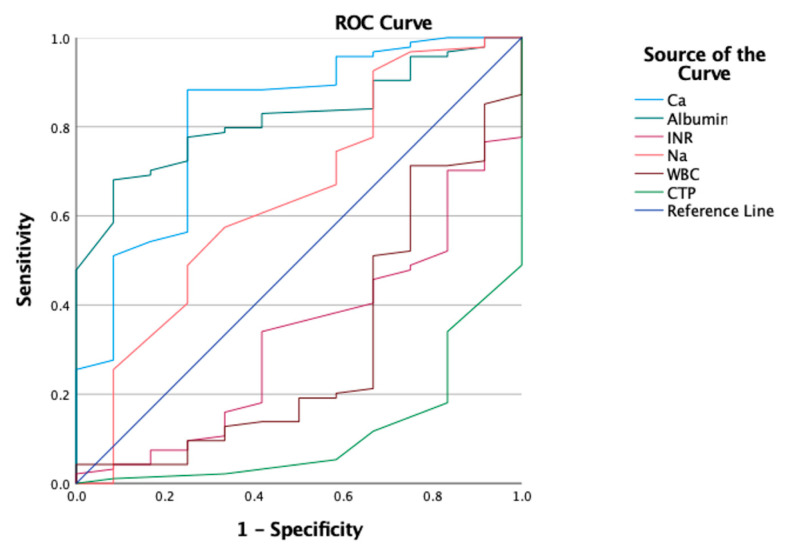
ROC curves of sodium, total serum calcium, albumin, INR, WBC and CTP. Na, sodium; INR, international normalized ratio; WBC, white blood cell; CTP, Child–Turcette-Pugh.

**Figure 5 biomedicines-11-01541-f005:**
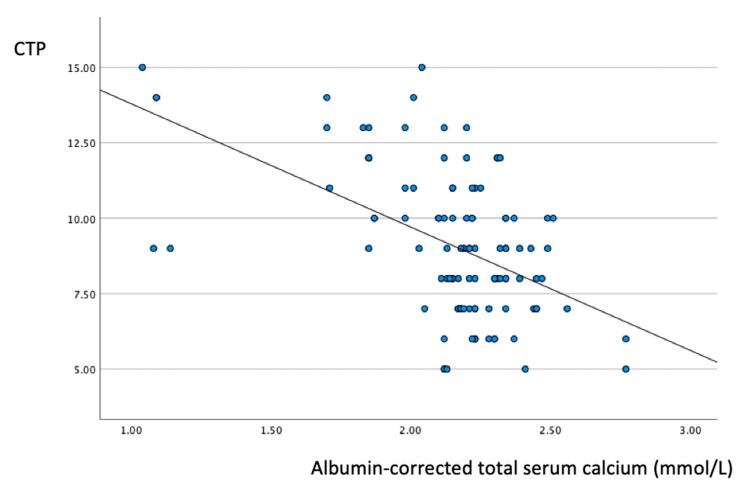
Scatter plot showing a significant negative correlation (r = −0.400; *p* < 0.05) between the total serum calcium values and CTP. CTP, Child–Turcette–Pugh.

**Table 1 biomedicines-11-01541-t001:** Comparison of clinical conditions and laboratory results on admission for COVID-19 patients with cirrhosis vs. COVID-19 patients without cirrhosis.

Variables	COVID-19 Patients without Cirrhosis (*n* = 399)	COVID-19 Patients with Cirrhosis (*n* = 52)	*p*
Age, years	65 (53–75)	62 (53–67)	**<0.05**
In-hospital mortality (%)	47 (11.8)	5 (9.6)	0.819
Hospital stay, days	11 (7–14)	14 (7–19)	0.082
Oxygen therapy (%)	147 (41)	21 (40.4)	0.649
Mechanical ventilation (%)	159 (39.8)	24 (46.2)	0.453
Cancer (%)	29 (7.3)	9 (17.3)	**<0.05**
Na, mmol/L	136 (132–138)	135 (130–138)	0.184
K, mmol/L	4.5 (4.12–4.9)	4.2 (3.7–4.6)	**<0.05**
Total serum calcium, mmol/L	2.08 (1.94–2.32)	1.44 (1.38–1.56)	**<0.05**
Corrected serum Ca *, mmol/L	2.32 (2.18–2.46)	2.16 (2.05–2.25)	**<0.05**
Mg, mmol/L	0.86 (0.79–0.94)	0.8 (0.74–0.86)	**<0.05**
AST, U/L	41 (29–62)	48 (31–102)	**<0.05**
ALT, U/L	31 (19–51)	35 (24–68)	0.109
Albumin, g/L	37 (32–43)	30.7 (26.8–35.2)	**<0.05**
INR	1.01 (0.87–1.17)	1.33 (1.11–1.53)	**<0.05**
Total Bilirubin, μmol/L	14.6 (8.5–21.4)	28.7 (13.1–73.8)	**<0.05**
Direct Bilirubin, μmol/L	6.87 (3.67–12.3)	9.6 (5.4–25.3)	**<0.05**
Total protein, g/L	65.4 (56.87–72.5)	63.9 (51.4–71)	0.051
GGT, U/L	51 (30–82)	86.5 (47–289)	**<0.05**
ALP, U/L	99 (78–170)	103 (78.5–205.8)	0.634
LDH, U/L	231 (167–312)	251 (213–304)	0.376
CRP, mg/L	114 (34.2–412)	62.2 (12.3–152.8)	**<0.05**
Hemoglobin, g/L	124 (102–130)	111.5 (102.3–131)	0.693
RBC, T/L	4.13 (3.78–4.65)	3.7 (3.35–4.06)	**<0.05**
WBC, G/L	8.03 (6.78–11.35)	10.0 (6–14.5)	0.111
Platelets, G/L	189 (123–312)	146 (95–221.8)	**<0.05**
Creatinine, μmol/L	92 (75–121)	115 (78.5–159.3)	**<0.05**
eGFR, mL/min	84 (67.8–90)	64.6 (42.8–90)	**<0.05**
Glucose, mmol/L	6.4 (5.76–7.46)	6.3 (5.6–6.8)	0.13
CTP score	-	9 (7–11)	**<0.05**
MELD-Na	-	20 (14–25)	**<0.05**

Statistically significant values are presented in bold. The data are presented as frequencies (%) or medians with IQRs. Na, sodium; K, potassium; Ca, calcium, Mg, magnesium; AST, aspartate aminotransferase; ALT, alanine aminotransferase; INR, international normalized ratio; GGT, gamma-glutamyl transferase; ALP, alkaline phosphatase; LDH, lactate dehydrogenase; CRP, C-reactive protein; RBC, red blood cell; WBC, white blood cell; eGFR, estimated glomerular filtration rate, CTP, Child–Turcette–Pugh; MELD-Na, Model for End-Stage Liver Disease sodium; IQR, interquartile range. * Corrected serum calcium represents the calcium level corrected for the serum albumin concentration.

**Table 2 biomedicines-11-01541-t002:** Univariate and multivariate regression analysis to identify prognostic factors for mortality.

Variable	Univariate Analysis				Multivariate Analysis			
	β	*p*	OR	95% CI	β	*p*	OR	95% CI
Age	0.080	**<0.05**	1.083	1.053–1.115	−0.090	**<0.05**	0.914	0.885–0.944
Hospital stay	0.152	**<0.05**	1.164	1.097–1.236	−0.138	**<0.05**	0.871	0.818–0.928
Na	−0.100	**<0.05**	0.905	0.847–0.966	0.096	**<0.05**	1.100	1.037–1.167
K	−0.020	0.938	0.980	0.588–1.634	0.224	0.352	1.251	0.781–2.007
Ca *	−0.574	0.418	0.563	0.141–2.257	1.583	**<0.05**	4.871	1.566–15.146
Mg	−1.594	0.276	0.203	0.012–3.572	2.424	0.084	11.292	0.724–176.050
AST	0.004	0.211	1.004	0.998–1.011	−0.005	0.216	0.995	0.987–1.003
ALT	−0.006	0.131	0.994	0.987–1.002	0.003	0.580	1.003	0.992–1.015
Albumin	−0.132	**<0.05**	0.876	0.827–0.929	0.143	**<0.05**	1.154	1.089–1.224
INR	1.259	**<0.05**	3.523	1.414–8.780	−1.356	**<0.05**	0.258	0.102–0.647
Total bilirubin	0.003	**<0.05**	1.003	1.000–1.006	−0.003	0.051	0.997	0.994–1.000
Direct bilirubin	−0.018	**<0.05**	0.982	0.968–0.997	0.014	**<0.05**	1.014	1.000–1.027
Total protein	−0.004	0.755	0.996	0.970–1.022	0.009	0.513	1.009	0.982–1.036
GGT	−0.001	0.395	0.999	0.997–1.001	0.000	0.698	1.000	0.998–1.002
ALP	0.001	0.329	1.001	0.999–1.002	0.000	0.592	1.000	0.998–1.001
LDH	−0.002	0.269	0.998	0.994–1.002	0.002	0.162	1.002	0.999–1.005
CRP	0.001	0.244	1.001	1.000–1.002	−0.001	0.069	0.999	0.998–1.000
Hemoglobin	−0.017	0.132	0.983	0.962–1.005	0.019	0.108	1.019	0.996–1.043
RBC	−0.591	0.068	0.554	0.293–1.045	0.622	0.065	1.863	0.963–3.604
WBC	0.273	**<0.05**	1.314	1.226–1.409	−0.300	**<0.05**	0.741	0.686–0.799
Platelets	−0.005	**<0.05**	0.995	0.990–1.000	0.005	0.057	1.005	1.000–1.010
Creatinine	0.002	0.331	1.002	0.998–1.006	−0.002	0.298	0.998	0.994–1.002
CTP	1.211	**<0.05**	3.358	1.545–7.300	−1.209	**<0.05**	0.299	0.115–0.775
MELD-Na	−0.128	0.635	0.880	0.519–1.493	0.264	0.490	1.301	0.616–2.750

Statistically significant values are presented in bold. Na, sodium; K, potassium; Ca, calcium, Mg, magnesium; AST, aspartate aminotransferase; ALT, alanine aminotransferase; INR, international normalized ratio; GGT, gamma-glutamyl transferase; ALP, alkaline phosphatase; LDH, lactate dehydrogenase; CRP, C-reactive protein; RBC, red blood cell; WBC, white blood cell; CTP, Child–Turcette–Pugh; MELD-Na, Model for End-Stage Liver Disease sodium; OR odds ratio; CI confidence interval. * Total serum calcium represents corrected calcium level for serum albumin concentration.

**Table 3 biomedicines-11-01541-t003:** Diagnostic efficacy of the six investigated laboratory parameters.

Prognostic Marker	AUC (95% CI)	Cut-Off	Sensitivity	Specificity	*p*
Na	0.643 (0.465–0.821)	133.5	0.745	0.417	0.108
Ca *	0.818 (0.683–0.953)	2.02	0.883	0.750	**<0.05**
Albumin	0.821 (0.729–0.914)	27.9	0.777	0.750	**<0.05**
INR	0.332 (0.181–0.482)	1.345	0.340	0.583	0.058
WBC	0.309 (0.136–0.481)	6.905	0.511	0.333	**<0.05**
CTP	0.115 (0.025–0.205)	9.5	0.340	0.167	**<0.05**

Statistically significant values are presented in bold. INR, international normalized ratio; WBC, white blood cell; CTP, Child–Turcette–Pugh; AUC area under the curve; CI confidence interval. * Corrected serum calcium represents the calcium level corrected for the serum albumin concentration.

**Table 4 biomedicines-11-01541-t004:** Comparison of different total serum calcium levels among COVID-19 patients with cirrhosis.

Variable	COVID-19 Patients with Cirrhosis (*n* = 52)		*p*
	Hypocalcemia (*n* = 25)	Normocalcemia (*n* = 27)	
Fatal outcome	4 (16)	1 (3.7)	0.183
Type of cirrhosis			0.430
compensated decompensated ACLF	1 (4) 20 (80) 4 (16)	3 (11.1) 22 (81.5) 2 (7.4)	
Age	65 (53–68)	60 (52–65)	0.359
Oxygen therapy	6 (24)	15 (55.6)	**<0.05**
Mechanical ventilation	16 (64)	8 (29.6)	**<0.05**
Ascites grades			**<0.05**
mild	0 (0)	2 (7.4)	
moderate	4 (16)	11 (40.7)	
severe	21 (84)	14 (51.9)	
Hospital stay	14 (5–20)	13 (8–17)	0.776
Albumin	28 (22.5–31.2)	32 (29–38)	**<0.05**
INR	1.36 (1.12–1.66)	1.2 (1.1–1.4)	0.148
Total bilirubin	32.1 (21–130)	24.1 (11.8–38.7)	0.200
Direct bilirubin	19.6 (8–26.1)	6.4 (4.1–14.1)	**<0.05**
Total protein	52 (45.8–65.4)	69 (61–73)	**<0.05**
GGT	149 (80–326)	64 (39–128)	**<0.05**
WBC	11.4 (6–16.9)	8.5 (5.9–14.5)	0.272
CRP	79.3 (15.2–122.8)	26.5 (3.9–176)	0.280
CTP	10 (9–12)	8 (7–10)	**<0.05**
MELD-Na	22 (15–25)	17 (14–23)	0.110

Categorical variables are presented as frequencies (percentages). Continuous variables are presented as medians (interquartile ranges). Bold text highlights the statistically significant values. ACLF, acute-on-chronic liver failure; INR, international normalized ratio; GGT, gamma-glutamyl transferase; WBC, white blood cell; CRP, C-reactive protein; CTP, Child–Turcette–Pugh; MELD-Na, Model for End-Stage Liver Disease sodium.

**Table 5 biomedicines-11-01541-t005:** Spearman’s correlation analysis between the total serum calcium and abnormal laboratory parameters.

Variables between	Spearman’s Correlation Coefficient	*p*
Ca * and Age	−0.139	0.325
Ca * and INR	−0.247	0.077
Ca * and Total bilirubin	−0.173	0.219
Ca * and Direct bilirubin	−0.275	**<0.05**
Ca * and GGT	−0.350	**<0.05**
Ca * and CRP	−0.341	**<0.05**
Ca * and WBC	−0.242	0.084
Ca * and CTP	−0.400	**<0.05**

A correlation is significant at the 0.05 level (*p* < 0.05). Bold text highlights the statistically significant values. R: Spearman’s correlation coefficient; INR, international normalized ratio; GGT, gamma-glutamyl transferase; WBC, white blood cell; CRP, C-reactive protein; CTP, Child–Turcette–Pugh. * Total serum calcium represents the calcium level corrected for the serum albumin concentration.

## Data Availability

Data are accessible upon reasonable request from the corresponding author.

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
