# Peer review of "Hypocalcemia on Admission Is a Predictor of Disease Progression in COVID-19 Patients with Cirrhosis: A Multicenter Study in Hungary"

_biomedicines, 2023, doi:10.3390/biomedicines11061541_

Round 1

Reviewer 1 Report

This is an interesting study on COVID 19 outcomes in patients with cirrhosis. I'm not convinced that the evidence presented clearly suggests a worse outcome in those with hypocalcemia. The main issue is that the authors have shown that patients with more advanced cirrhosis have lower Ca levels. However, when they split the cirrhotic group into low and normal Ca, the mortality rates are similar.

Selection: The authors need to explain more clearly how they selected their study cohort out of the 6394 patients with COVID.

Table 1: there is a space between AST and ALT which pushes all the results down the page and makes it difficult to interpret.

In section 3.4, the authors talk about low Ca being associated with disease progression but I don't think they show any longitudinal results. They don't describe how patients became worse. Do they mean that Ca levels are associated with disease severity?

Figure 1 does not add anything to the manuscript since the p values in non-significant.

Reviewer 2 Report

I read with interest paper entitled “Hypocalcaemia on admission is a predictor of poor prognosis in COVID-19 patients with cirrhosis: a multicentre study in Hungary”. The authors aimed to investigate the association of hypocalcaemia with COVID-19 severity in patients with liver cirrhosis. Although interesting, there are some issues that need to be addressed.

Major issues:

1.)   METHODOLOGY:

a.     Inclusion and exclusion criteria should be clearly stated.

b.     could you clarify why did the study only include pts with elevated liver transaminases in both study groups? 

c.      Were there any pts with cirrhosis without liver transaminases >40 U/L? If so, were they excluded and why?

d.     HRCT was performed on admission for the initial diagnosis of COVID-19 pneumonia – was this done for all of the included patients? How many patients were there without pneumonia? It is not clear if the study included only pts with pneumonia or just the hospitalized sars-cov-2 patients aged ≥ 18 years with liver transaminases >40 U/L, regardless of COVID-19 severity or presence of pneumonia – e.g.  were pts without requirement of supplemental oxygen included as were mechanically ventilated pts?

e.     Could the comparison of non-albumin-corrected calcium levels give us further insight on absolute calcium deficiency impact on outcomes, since only the albumin-corrected-calcium was compared between the groups. I.e. could hypocalcemia solely due to hypoalbuminemia have a statistically significant impact on outcomes?

f.      Could it be useful to show data and a commentary for COVID-19 pts with and without hypocalcaemia, as was shown in Table 3 for pts with cirrhosis?

g.     Etiology and severity of liver cirrhosis should be included in results. If MELD-Na was calculated from admission laboratory data, this could be important cofounder and stated in limitations.

h.     When assessing liver cirrhosis and mortality data on time since first decompensation and recent hospitalizations due to liver cirrhosis should be provided.

2.)    LIMITATIONS:

a.     Were there any immunocompromised pts or were they excluded?

b.     Did the study take into account blood pH levels of the pts, since it can vary considerably in respiratory infections like COVID-19 and considerably affect calcium binding?

c.      If two negative RT-PCR tests in a row, at least 24 hours apart were required for hospital discharge hospital stay and in-hospital mortality could vary considerably, not influenced by the clinical and/or laboratory criteria for hospital discharge.

d.     Important clinical parameters were not shown – type of respiratory support, in-hospital complications such as hospital-acquired infections, pulmonary thrombosis etc.

e.     The only outcome evaluated were duration of hospitalization and in-hospital mortality

3.)   RESULTS:

a.     Since data are non-normally distributed, medians with IQR should be used instead of mean with SD.

b.     Describe the multivariate model used

MINOR ISSUES:

4.)   Appearing multiple times throughout the paper:

a.     till  until;

b.     COVID-19 infection  sars-cov-2 infection or just COVID-19;

c.      international data  data;

d.     following covid  with covid OR following sars-cov-2 infection;

e.     maybe better to exclude “to date”

f.      maybe to exclude “international” from “international data”

g.     row 238 and 239: Child-Turcette-Pugh can be replaced by CTP

5.)   Row 17: and IS associated OR cirrhosis WHICH IS associated

6.)   Row 20: disease  COVID-19

7.)   Row 20 and 60: – maybe it should be specified the association of which abnormal laboratory parameters with hypocalcemia was one of the aims?

8.)   Row 39: deterioration in cirrhosis  progression of cirrhosis i.e. that covid-19 is associated with the risk of cirrhosis progression and poor outcome

9.)   Row 42: There are also an increased gut permeability and intestinal dysbiosis, which lead to increased bacterial translocation and consequently the increased  Furthermore, this condition is associated with increased bacterial translocation and increased prevalence…

10.) Row 47: In the immune cells  exclude “the”

11.) Can virulence of a pathogen be associated with host factors?

12.) To date, several studies have shown the relation between lower levels of serum calcium and disease severity in COVID-19 infection.  have shown…between the level of serum calcium…

13.)  Row 92: saturation ≥ 94%  without supplementary oxygen therapy

14.)  Row 94: two negative RT-PCR tests in a row  specify on which sample.

15.)  Row 98: there is no need to specify the components of the complete blood count here.

16.)  Row 118: maybe to change CLIF to EASL-CLIF

17.)  Row 119: …accordance with the EASL-CLIF…

18.)  Row 123: No need to explain again what “INR” stands for

19.)  Row 152: It says the mean value of CRP was significantly higher in cirrhosis patients which is in contrast to the information in Table 1.

20.)  Table 1.:

a.     include units of measurement for laboratory parameters

b.     The median CRP value in pts without cirrhosis is surprisingly high – this should be commented on. Also, it is peculiar for patients with transaminases <40 U/L (i.e., without COVID-associated liver injury which is an inclusion criterion) to have such high levels of CRP.

c.      How were p-values calculated for CTP and MELD-Na?

21.)  Row 156 and 176: Na=natrium  sodium

22.)  Row 185: it should be emphasized that the mortality difference was not statistically significant.

23.)  Row 187: no need to explain what ACLF stands for again.

24.)  Row 188: was this statistically significant, or should this sentence be excluded given the low numbers?

25.)  Rows 205-208 should be deleted (repetition of rows 200-202, and repetitive explanation of the abbreviation ACLF)

26.)  Row 231: the mentioned studies do not state that two negative PCR tests were required for discharge.

27.)  Rows 238 and 239: Child-Turcette-Pugh score  CTP

28.)  Rows 246 and 284: hypocalcaemia on admission was a significant predictor for mortality in COVID-19 patients with cirrhosis  was it? Has the study shown this significant association?

29.)  Row 247: calcium is not a trace element

30.)  Row 253: Virus reproduction  replication

31.)  Row 256: Is there any other parameters od COVID-19 severity which can be stated here besides levels of CRP?

32.)  Row 261: It should be clarified how is COVID-19 severity defined here and is does this study provide the evidence on this association or does it just suggest this association pending further research.

Round 2

Reviewer 2 Report

The authors have sufficiently responded to all my comments and significantly improved the manuscript.

I would suggest improving the formatting of Table 1 (exclude medinas and IQR from column 1; it is unnecessary to repeat measurement units in other columns; for example:

Age, years       65 (53-75)       62 (53-67)       < 0.05

And in footnote you can add * the data are presented as frequencies (%) or medians with IQR.

I do not have any further complaints.
